# Test-Time Distribution Normalization
# for Contrastively Learned Vision-language Models

**Yifei Zhou**[*]
University of California, Berkeley
yifei_zhou@berkeley.edu

**Juntao Ren**[*]
Cornell University
jlr429@cornell.edu

**Fengyu Li**[*]
Cornell University
fl334@cornell.edu

**Ramin Zabih**
Cornell University
rdz@cs.cornell.edu

**Ser-Nam Lim**
University of Central Florida
sernam@ucf.edu

## Abstract

Advances in the field of vision-language contrastive learning have made it possible for many downstream applications to be carried out efficiently and accurately by simply taking the dot product between image and text representations. One of the most representative approaches proposed recently known as CLIP [50] has garnered widespread adoption due to its effectiveness. CLIP is trained with an InfoNCE loss that takes into account both positive and negative samples to help learn a much more robust representation space. This paper reveals that the common downstream practice of taking a dot product is only a zeroth-order approximation of the optimization goal, resulting in a loss of information during test-time. Intuitively, since the model has been optimized based on the InfoNCE loss, test-time procedures should also be in alignment. The question lies in how one can retrieve any semblance of negative samples information during inference in a computationally efficient way. To this end, we propose Distribution Normalization (DN), where we approximate the mean representation of a batch of test samples and use such a mean to represent what would be analogous to negative samples in the InfoNCE loss. DN requires no retraining or fine-tuning and can be effortlessly applied during inference. Extensive experiments on a wide variety of downstream tasks exhibit a clear advantage of DN over the dot product on top of other existing test-time augmentation methods. Our code is available at https://github.com/fengyuli2002/distribution-normalization.

## 1   Introduction

Contrastive Learning [20, 3, 60, 6, 52] has become an important area of research in recent years due to its widespread success in self-supervised representation learning. The key idea behind contrastive learning is to group positive samples that share similar semantics (e.g., images resulting from applying different augmentations to the same image), while separating semantically different negative samples (e.g., different images with different data augmentations). On the heels of this progress, contrastive learning has also been shown to be very effective for creating a joint representation space between images and text [50, 46, 75, 10]. The CLIP approach proposed in [50] is at the forefront of this research effort and has been shown to be the state-of-the-art in multiple downstream tasks. The model is generally pre-trained on a large amount of image-text pairs that form the positive samples, while the negative samples are composed from random pairing of images and text. After the model is trained, similarities between text and images can be measured by simply taking a dot product between their corresponding representations. Such a paradigm has opened the doors to state-of-the-art performance

---

[*]Equal Contribution

37th Conference on Neural Information Processing Systems (NeurIPS 2023).

in many cross-modal alignment tasks such as zero-shot classification [73, 21], cross-modal retrieval [71, 40, 35, 32], and evaluation of machine-generated captions [17, 22, 30].

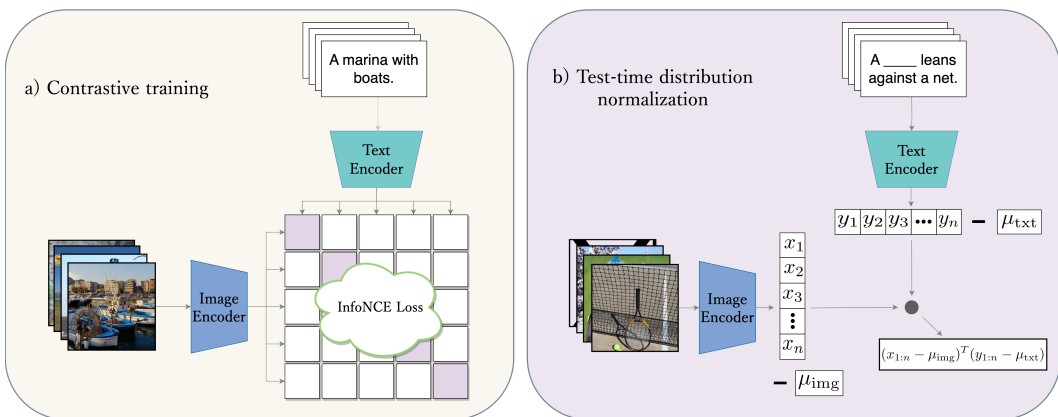

Figure 1: An illustration of how Distribution Normalization better aligns with infoNCE pre-training objective. $\mu_{img}$ and $\mu_{txt}$ are mean encoded representations for images and text respectively.

Despite CLIP's successes, we observe a "mismatch" between the pre-training objective of such cross-modal representation models and their downstream usage. Most if not all of these models are trained taking into account of both positive and negative samples, of which one of the most popular optimization objectives is the InfoNCE loss [58, 62, 74]. The InfoNCE loss considers negative samples from the data distribution per positive sample and has been shown to be very effective for cross-modal representation learning. In contrast, during test-time, the similarity is often measured using the dot product between the image and the text representations, which does not utilize any information with regard to the data distribution.

This paper presents our findings that rectifying such misalignment can boost performance consistently across a variety of downstream tasks. However, naively applying InfoNCE loss for downstream tasks requires iterating over all negative samples for every test sample, which is not a tractable operation. Instead, we found that a first-order approximation of the InfoNCE loss, consisting of simply subtracting the distribution mean in the representation space before taking the dot product, is able to achieve a similar effect. We call this proposed approach Distribution Normalization (DN) – DN is very easy to implement and does not require any retraining, fine-tuning or labeled data.

To investigate how this simple modification affects the performance of CLIP and its state-of-the-art variants in practice, we conduct extensive experiments on a wide variety of cross-modal alignment tasks, including zero-shot classification, cross-modal retrieval, and evaluation of image captions. In all the settings that we have studied, DN displays a clear benefit over direct dot product and can be combined with other existing test-time augmentation methods [55], which we will show in our experiments. Our contributions are: (1) We present an insightful analysis of why the dot product as a similarity measure is only a zeroth-order approximation; (2) We propose DN to overcome this shortcoming by utilizing an approximation of the test-time distribution, and also provide guidance on how DN can be conveniently implemented in practice; (3) We provide extensive supporting empirical evidence and ablation studies that demonstrate the fidelity of DN for practical use.

## 2 Background & Related Works

**Learning via Natural Language Supervision** Several recent studies [7, 12, 34, 35] leverage large-scale datasets of image-text pairs to supervise vision-language pre-training (VLP) with a contrastive objective. CLIP [50] is one such example that is contrastively trained to create a joint vision-language representation space, which has produced empirical success in downstream tasks such as zero-shot classification and cross-modal retrieval over previous state-of-the-art models [40, 65, 68, 33]. Since CLIP's introduction, several other works have utilized a similar contrastive objective to create good vision-language representations and achieve state-of-the-art performance on many downstream tasks involving datasets that the model was not pre-trained on [33, 21, 7, 11, 26]. For example, ALIGN scales joint pre-training to minimally-filtered datasets [21], while ALBEF [33] and TCL [68] focus on learning more semantically meaningful multi-modal interactions. It is noteworthy that these SOTA

models all leverage the InfoNCE loss as their key training objective, and apply a simple dot product in downstream tasks.

**CLIP Augmentation/Adaptation for Downstream Applications** Researchers have explored numerous methods to adapt CLIP for downstream tasks, capitalizing on its remarkable transferability. CoOp [75, 76] employs learnable textual prompts, inspired by prompt learning [36], for few-shot training data, while VT-CLIP [49] incorporates visual-guided texts for enhanced vision-language alignment. However, these adaptations demand extra training data and fine-tuning resources, undermining CLIP's core strength of efficient zero-shot recognition. To address this issue, CALIP [14] presents a training-free attention module to boost zero-shot image classification, and SuS-X [61] offers training-free name-only transfer by adapting CLIP with a curated support set. However, both CALIP and Sus-X are tailored for zeroshot classification only, while in this paper, we propose DN, a training-free adaptation that is conceptually simpler, easier to implement, and applicable to a wide range of downstream tasks.

**Test-time Augmentation/Adaptation** Test-time adaptation and test-time augmentation strategies share a common goal to adapt to out-of-distribution scenarios [53, 43] during test time. Many test-time adaptation techniques make adjustments to the batch normalization layers to normalize feature statistics [37, 41], while test-time augmentation techniques [64, 25, 42, 29, 15, 19, 57] aim to ameliorate performance by taking some aggregation [55] of the model's prediction over a batch of augmented images. Though DN is also applied during test-time, DN's task is to align the mismatch between training and inference objectives *and not* to deal with out of distribution test samples. This renders DN orthogonal to TTA. In our experiments, we use a representative TTA model from Shanmugam et al. [55] and show that while TTA improves model performance, adding DN to TTA gives an additional increase.

## 3    Methodology

This section is organized as follows: in Section 3.2, we show that a dot product is only a zeroth-order approximation of the pre-training objective and how a better approximation arises naturally from the InfoNCE objective, and in Section 3.3, we explain how this idea can be efficiently implemented in practice.

### 3.1    Notations

In our analysis, $\mathcal{D}_S$ is the training distribution and $\mathcal{D}_T$ is the test distribution, $x_0, y_0$ are images and text belonging to the same pair, $x_1, ..., x_n$ and $y_1, ..., y_n$ are random images and text in the same batch, $\phi, \psi$ are image and text encoders respectively, and $\tau$ is a temperature hyperparameter used in InfoNCE loss while training. Since in most applications, only relative comparisons of the distances matter (e.g., retrieving the best candidate caption describing the image), we consider two similarity measures $S_1$ and $S_2$ to be equivalent if one is a monotone function of the other, namely

$$\forall x_1, y_1, x_2, y_2, S_1(x_1, y_1) \geq S_1(x_2, y_2) \Leftrightarrow S_2(x_1, y_1) \geq S_2(x_2, y_2). \tag{1}$$

The goal here is to find a test-time similarity measure that is similar to the train-time optimization goal through careful simplification of InfoNCE loss.

### 3.2    Analysis of InfoNCE Loss

We start from the original form of InfoNCE loss [58, 62, 74]:

$$\mathcal{L}_{NCE}(\mathcal{D}_S) = \mathbb{E}_{x_0, y_0 \sim \mathcal{D}_S}[\mathbb{E}_{y_1, ..., y_n \sim \mathcal{D}_S} \log \frac{1}{1 + \frac{\sum_{i=1}^n e^{\phi(x_0)^\intercal \psi(y_i)/\tau}}{e^{\phi(x_0)^\intercal \psi(y_0)/\tau}}} + \mathbb{E}_{x_1, ..., x_n \sim \mathcal{D}_S} \log \frac{1}{1 + \frac{\sum_{i=1}^n e^{\phi(x_i)^\intercal \psi(y_0)/\tau}}{e^{\phi(x_0)^\intercal \psi(y_0)/\tau}}}].$$

$$\tag{2}$$

Since $\frac{\sum_{i=1}^n e^{\phi(x_i)^\intercal \psi(y_0)/\tau}}{e^{\phi(x_0)^\intercal \psi(y_0)/\tau}}$ is usually quite small ($< 0.1$) due to the exponentiation, we can taylor expand the above term around 0 and discard second and higher order terms, resulting in

$$\mathcal{L}_{NCE}(\mathcal{D}_S) \approx \mathbb{E}_{x_0, y_0 \sim \mathcal{D}_S}[\mathbb{E}_{y_1, ..., y_n \sim \mathcal{D}_S} \frac{\sum_{i=1}^n e^{\phi(x_0)^\intercal \psi(y_i)/\tau}}{e^{\phi(x_0)^\intercal \psi(y_0)/\tau}} + \mathbb{E}_{x_1, ..., x_n \sim \mathcal{D}_S} \frac{\sum_{i=1}^n e^{\phi(x_i)^\intercal \psi(y_0)/\tau}}{e^{\phi(x_0)^\intercal \psi(y_0)/\tau}}],$$

$$\tag{3}$$

which can be simplified to

$$\mathcal{L}_{NCE}(\mathcal{D}_S) \approx n\mathbb{E}_{x_0,y_0 \sim \mathcal{D}_S}[\mathbb{E}_{y_1 \sim \mathcal{D}_S} e^{\phi(x_0)^\intercal[\psi(y_1)-\psi(y_0)]/\tau} + \mathbb{E}_{x_1 \sim \mathcal{D}_S} e^{[\phi(x_1)-\phi(x_0)]^\intercal \psi(y_0)/\tau}]. \quad (4)$$

A successful generalization to the test distribution means that the infoNCE loss $\mathcal{L}_{NCE}(\mathcal{D}_T)$ is small when the expectation over training distribution for $x_0, y_0, x_1, y_1$ is substituted with the expectation over test distribution, where

$$\mathcal{L}_{NCE}(\mathcal{D}_T) \approx n\mathbb{E}_{x_0,y_0 \sim \mathcal{D}_S}[\mathbb{E}_{y_1 \sim \mathcal{D}_T} e^{\phi(x_0)^\intercal[\psi(y_1)-\psi(y_0)]/\tau} + \mathbb{E}_{x_1 \sim \mathcal{D}_T} e^{[\phi(x_1)-\phi(x_0)]^\intercal \psi(y_0)/\tau}]. \quad (5)$$

### 3.2.1 Reduction to Zero-Order Dot Product

We argue that Eqn. 5 can be reduced to straightforwardly comparing the dot products. When we do not have of the test distribution $\mathcal{D}_T$, the only thing that we can do is to perform a zeroth-order approximation of Eqn. 5. The most intuitive way of doing so is to reduce the distribution of $\phi(x_1)$ and $\psi(y_1)$ to be 0 deterministically, so that:

$$\mathcal{L}_{NCE}^{(0)}(\mathcal{D}_T) = 2n\mathbb{E}_{x_0,y_0 \sim \mathcal{D}_T} e^{-\phi(x_0)^\intercal \psi(y_0)/\tau}. \quad (6)$$

This shows that we can essentially just output $e^{-\phi(x_0)^\intercal \psi(y_0)}$ as the distance between $x_0$ and $y_0$. So we get the zeroth-order similarity:

$$S_{(0)}(x_0, y_0) = \phi(x_0)^\top \psi(y_0). \quad (7)$$

### 3.2.2 Efficient First-Order Approximation

The common practice of taking a naive zeroth-order approximation can result in a loss of some important information of the distribution of negative samples. On the other hand, computing Eqn.5 for each pair $x_0, y_0$ involves iterating over all samples in a distribution $\mathcal{D}_T$, and is computationally inefficient. We therefore turn to more efficient approximations using statistics of higher-order moments of the distribution. In our experiments, we found that this intuitive modification preserves most of the useful information as shown in Appendix D.2.

If we can estimate the first moment of the distribution, i.e., the mean of the unlabeled distribution $\mu_x$ and $\mu_y$ for $\phi(x_1)$ and $\psi(y_1)$, we can perform a first-order approximation of the distribution with $\widehat{P}(\phi(x_1)) = \mathbb{I}\{\phi(x_1) = \mu_x\}$ and $\widehat{P}(\psi(y_1)) = \mathbb{I}\{\psi(y_1) = \mu_y\}$, so that $\widehat{P}(\phi(x_1)), \widehat{P}(\psi(y_1))$ matches the true distribution in terms of the first-order moment. This simplifies Eqn. 5 to be:

$$\mathcal{L}_{NCE}^{(1)}(\mathcal{D}_T) = \mathbb{E}_{x_0,y_0 \sim \mathcal{D}_T}[e^{\phi(x_0)^\intercal(\mu_y-\psi(y_0))/\tau} + e^{(\mu_x-\phi(x_0))^\intercal \psi(y_0)/\tau}]. \quad (8)$$

Therefore, to evaluate the distance of any $x_0, y_0$, we can simply output $e^{\phi(x_0)^\intercal(\mu_y-\psi(y_0))/\tau} + e^{(\mu_x-\phi(x_0))^\intercal \psi(y_0)/\tau}$ without having to calculate an expectation by iterating through all samples in the distribution.

## 3.3 Distribution Normalization

Considering that $e^{\phi(x_0)^\intercal(\mu_y-\psi(y_0))/\tau} + e^{(\mu_x-\phi(x_0))^\intercal \psi(y_0)/\tau}$ is equivalent to taking an algorithmic mean of $e^{\phi(x_0)^\intercal(\mu_y-\psi(y_0))/\tau}$ and $e^{(\mu_x-\phi(x_0))^\intercal \psi(y_0)/\tau}$, to simplify this further, we can convert this algorithmic mean to a geometric mean $\sqrt{e^{\phi(x_0)^\intercal(\mu_y-\psi(y_0))/\tau} e^{(\mu_x-\phi(x_0))^\intercal \psi(y_0)/\tau}}$ which achieves a similar effect as averaging the distance measure of $e^{\phi(x_0)^\intercal(\mu_y-\psi(y_0))/\tau}$ and $e^{(\mu_x-\phi(x_0))^\intercal \psi(y_0)/\tau}$. An additional empirical verification of the correlation between algorithmic mean and geometric mean in this step is provided in Appendix D.2. Since $\sqrt{e^{\phi(x_0)^\intercal(\mu_y-\psi(y_0))/\tau} e^{(\mu_x-\phi(x_0))^\intercal \psi(y_0)/\tau}}$ is a monotone function of $-(\phi(x_0) - \frac{1}{2}\mu_x)^\intercal(\psi(y_0) - \frac{1}{2}\mu_y)$ after removing the square root and exponentiation, we can simply use $-(\phi(x_0) - \frac{1}{2}\mu_x)^\intercal(\psi(y_0) - \frac{1}{2}\mu_y)$ to calculate the distance, resulting in a simplified first-order similarity measure:

$$S_{(1)}(x_0, y_0) = (\phi(x_0) - \frac{1}{2}\mu_x)^\intercal(\psi(y_0) - \frac{1}{2}\mu_y). \quad (9)$$

Note that the new similarity measure is equivalent to subtracting the means immediately after the original representations are calculated, a method we call Distribution Normalization (DN), after

which the similarity is just their dot product. Clearly, our proposal is extremely easy to implement when access to the mean of the distribution of interest is available.

Fortunately, we observed that the mean of the distribution can usually be robustly estimated from very few unlabeled samples as shown in Section 4.6.3. This is easily accessible in all the downstream tasks that we have considered in this paper and is much cheaper than fine-tuning where not only more data is required but also the data needs to be annotated. For example, in the task of cross-modal retrieval, the mean can be estimated from the gallery data and past queries (without knowing the correct answer to the queries); in the task of zero-shot classification, the mean can be estimated from a small and unlabeled validation set; and in the task of image caption metrics, the mean can be estimated from the images to be captioned and the references provided.

# 4 Experiments

Our experiments are designed to answer the following questions: 1) whether our proposed DN can uniformly improve a wide range of cross-modal alignment tasks for different kinds of cross-modal representation models and whether this gain is larger than that achieved by other zeroshot CLIP augmentations, 2) whether DN can be used in parallel with other common test-time adaptation methods compatible with CLIP, 3) how robust DN is when only scarce, unlabeled data is available to estimate the mean from, and 4) whether DN can improve the performance of fine-tuned models in addition to pre-trained models.

## 4.1 Downstream Tasks

**Cross-modal Retrieval** includes two subtasks where images are used to query corresponding text and vice versa. Following [68, 33], for each subtask, we consider both the zero-shot setting and the fine-tuning setting on COCO [39] and Flickr30K [48]. In the zero-shot setting, we directly apply the pre-trained model on the test set of each dataset while in the fine-tuned setting we first fine-tune each pre-trained model on the training set of each dataset with the InfoNCE loss and apply the fine-tuned model to the test set. For each setting, we report Recall@k (k = 1,5) whereby the retrieval is considered a success if its corresponding image/text is in the top-k.

**Zero-shot Classification** is the task where we directly apply the pre-trained models to predict the category of the images in the test set. Each class is represented with a fixed textual template of "a photo of {class_name}.", and the corresponding representation is obtained by encoding those sentences with the text encoder of the pre-trained model. For each image, we encode it with the image encoder and output the top-k most similar class representations as predictions. We benchmark all methods with top-1 and top-5 accuracy on ImageNet1K [9], Cifar100 [28], SUN397 [67], Stanford Cars [27], Caltech 101 [31], and Flowers102 [44] datasets.

**Image Captioning Metric** is the task that measures the correlation between human ratings and similarity metrics on machine-generated captions. Following [17], we consider both the reference-free setting and the reference-based setting. In the reference-free setting, the metric does not have access to human-written example captions (references) and has to give a score based on the source image and the generated caption alone while in the reference-based setting the metric can additionally take advantage of the information from the provided references.

## 4.2 Baselines

This paper compares the augmentations of Distribution Normalization and other baselines on CLIP [50], ALBEF [33], and TCL [68]. More details of the base representation models can be found in Appendix B. In particular, the following are the main baselines being examined in the experiments.

**TTA** [55] aims to better adapt vision encoders to out-of-distribution test samples by augmenting test images and taking a weighted reduction of the various augmentations. However, since DN is meant for the zero-shot setting, training these test-time augmenting layers is not applicable in our setting. Instead, we apply the standard test-time augmentation policy from Shanmugam et al. [55] which consists of a combination of flips, crops and scales, and take a mean over the model outputs on these augmented images.

**CALIP** [14] replaces CLIP's dot-product similarity with a parameter-free attention module. It improves performance on zero-shot image classification by enabling visual and textual representations

to better explore cross-modal features. This method results in improved adaptive zero-shot alignment by blending images with textual-aware signals. We implement CALIP using their official code and report our results on the "ViT-B/32"-based CLIP architecture, consistent with our approach in other experiments. Since the original paper focuses solely on zero-shot image classification, CALIP serves as our baseline for this task.

**TPT** [56] is another CLIP zeroshot method that performs prompt tuning on each single test image to promote the model's prediction consistency on different augmented views of the single test image. TPT is mainly proposed for classification tasks and it is unclear how it can be used for general cross-modal alignment tasks. We implement TPT using their official code and report our results on the "ViT-B/32"-based CLIP architecture, consistent with our approach in other experiments. Importantly, TPT is much more computationally demanding than vanilla CLIP dot product because it requires taking gradient steps for each single test image over its 64 different augmentations, while other methods do not suffer from such computation issues.

**DN** is our proposed plug-in augmentation using Eqn.9. Estimated image mean and text mean are respectively subtracted from image and text embeddings immediately after the embeddings are calculated from base encoder models. Dot products are then taken between text and image representations as the similarity. Different ways to estimate the image mean and text mean from few validation samples or references in different downstream tasks are detailed in the following sections.

**DN\*** Considering the motivation for DN is to better align the downstream similarity with the pre-training goal, it might introduce instability when there is a significant mismatch between the pre-raining and testing distribution. We therefore examine this variant called DN\* where the only difference from DN is that DN\* averages the similarity output by DN and a vanilla zero-order dot product for improved robustness.

## 4.3 Cross-modal Retrieval Results

We conducted experiments on both image-to-text and text-to-image retrieval on Microsoft COCO [39] and Flicker30K [48] (see Appendix C.1 for datasets details). For all the retrieval tasks, we estimated the mean with 100 random unlabeled samples from the validation set and calculated standard deviations and average recalls with 5 random seeds.

The results for both datasets are presented in Table 1 for the zero-shot setting. For all three base models, adding DN almost always improves recall for both zero-shot image-to-text and text-to-image retrieval tasks, with an average 1.1% increase in top-1 recall. This provides substantial evidence that DN is effective for both tasks even in datasets with diverse images like MSCOCO. The improvement can be as large as a 2.2% increase in top-1 image-to-text recall as we see for CLIP on Flickr30K. We also observe that, in general, adding DN brings a larger improvement for top-1 recall than for top-5/10 recall, suggesting that DN is particularly suitable for distinguishing similar samples.

While adding test-time augmentation techniques to CLIP are effective in themselves as shown in Table 1, we also show that adding DN or DN\* on top of TTA techniques further improves retrieval accuracy. Specifically, CLIP+TTA+DN achieves another average 1.3% improvement over CLIP+TTA. A similar improvement is achieved by CLIP+TTA+DN\* as well. However, it is noteworthy that although CLIP+TTA+DN and CLIP+TTA+DN\* achieves similar average improvement over CLIP + TTA (1.3% and 1.2% respectively), the improvement of CLIP+TTA+DN\* has a smaller standard deviation (0.8% compared to 1.3%), suggesting that averaging DN with direct dot product indeed improves the stability.

## 4.4 Zero-shot Classification Results

Table 2 gives the zero-shot classification results for all three base models and their variants with DN on ImageNet1K [9], Cifar100 [28], SUN397 [67], Stanford Cars [27], Caltech 101 [31], and Flowers 102 [44] (see Appendix C.2 for datasets details). In these experiments, we perform DN before the final layer normalization of image/text encoders.

DN consistently enhanced the performance of all three base models across six datasets with an average 4.4% increase in top-1 accuracy. It demonstrates effectiveness not just in general-purpose vision datasets like ImageNet, but also in specialized datasets such as Stanford Cars, where average accuracy increases by 1.5%. Specifically, CLIP+TTA+DN/DN\* is the best variant for a CLIP base model, signifying simultaneous benefits from DN and TTA. While CLIP+TTA+DN and CLIP+TTA+DN\*

| | MSCOCO (5K test set) | | | | | | Flickr30K (1K test set) | | | | | |
| | Image → Text | | | Text → Image | | | Image → Text | | | Text → Image | | |
| | R@1 | R@5 | R@10 | R@1 | R@5 | R@10 | R@1 | R@5 | R@10 | R@1 | R@5 | R@10 |
|---|---|---|---|---|---|---|---|---|---|---|---|---|
| CLIP [50] | 52.4 | 76.0 | 84.5 | 30.2 | 55.1 | 66.4 | 81.3 | 95.0 | 98.5 | 62.7 | 86.0 | 92.0 |
| CLIP + TTA [55] | 53.9 | 77.5 | 85.5 | 32.1 | 57.5 | 68.3 | 83.2 | 96.8 | 98.4 | 65.2 | 87.9 | 92.9 |
| CLIP + TTA + DN | 53.6±0.1 | 76.9±0.1 | 84.8±0.1 | **34.8**±0.0 | **60.4**±0.0 | **70.8**±0.1 | 85.8±0.2 | **97.5**±0.1 | **99.1**±0.0 | **68.1**±0.1 | **89.4**±0.1 | **94.1**±0.0 |
| CLIP + TTA + DN* | **54.7**±0.1 | **77.8**±0.1 | **85.6**±0.1 | 33.8±0.0 | 59.4±0.0 | 70.1±0.0 | 85.8±0.1 | **97.5**±0.1 | 98.8±0.1 | 67.6±0.0 | 89.1±0.0 | 93.9±0.1 |
| CLIP + DN | 51.7±0.1 | 75.8±0.1 | 84.0±0.1 | 33.4±0.0 | 58.6±0.1 | 69.4±0.1 | 83.3±0.2 | 96.4±0.1 | 98.6±0.1 | 66.2±0.1 | 88.2±0.1 | 93.3±0.1 |
| CLIP + DN* | 52.9±0.1 | 76.4±0.1 | 84.9±0.1 | 32.1±0.1 | 57.4±0.0 | 68.3±0.1 | 83.5±0.1 | 96.2±0.0 | 98.5±0.1 | 64.8±0.2 | 87.5±0.1 | 93.1±0.0 |
| TCL [68] | 57.6 | 84.3 | 91.8 | 41.8 | 70.6 | 80.6 | 73.8 | 93.3 | 96.9 | 59.1 | 84.6 | 91.1 |
| TCL + DN | **60.6**±0.1 | **85.8**±0.1 | **92.4**±0.1 | **43.2**±0.0 | **71.8**±0.1 | **81.6**±0.1 | 77.5±0.5 | 94.1±0.2 | 96.9±0.2 | 59.8±0.2 | 84.9±0.1 | 91.1±0.1 |
| TCL + DN* | 59.5±0.1 | 85.2±0.0 | 92.2±0.1 | 42.7±0.0 | 71.5±0.0 | 81.3±0.0 | 75.5±0.0 | **94.4**±0.1 | 96.9±0.1 | **60.0**±0.1 | **85.1**±0.0 | 91.1±0.0 |
| ALBEF [33] | 62.5 | 85.9 | 92.2 | 40.2 | 68.4 | 78.9 | 78.2 | 95.5 | 97.9 | 59.9 | 84.8 | 90.6 |
| ALBEF + DN | **63.0**±0.2 | 85.8±0.1 | 92.4±0.1 | **44.8**±0.1 | **72.5**±0.1 | **82.0**±0.0 | **80.6**±0.1 | **96.2**±0.1 | **98.3**±0.1 | **64.1**±0.0 | **87.1**±0.1 | **92.3**±0.1 |
| ALBEF +DN* | **63.0**±0.1 | **86.0**±0.0 | **92.5**±0.1 | 42.8±0.1 | 70.8±0.0 | 80.7±0.0 | 79.2±0.1 | **96.2**±0.0 | 98.0±0.0 | 62.4±0.1 | 86.1±0.1 | 91.9±0.1 |

Table 1: Cross-modal retrieval performance on MSCOCO and Flickr30K in the zero-shot setting. Means for DN are estimated using 100 random unlabeled validation samples. Average recalls and standard deviations are calculated with 5 random seeds.

display similar improvements over CLIP+TTA (0.82% and 0.9% respectively), CLIP+TTA+DN* again exhibits a smaller standard deviation (0.22% vs. 0.76%), indicating improved stability. Furthermore, DN consistently surpasses CALIP in all model-dataset combinations, despite CALIP's slight higher computational cost from incorporating an attention module. Comparing TPT with our best variant CLIP+TTA+DN*, we find that they achieve comparable top-1 accuracy across all datasets while TPT suffers from a significant drop in top-5 accuracy, potentially due to poor calibrations in performing gradient steps on each single test image. It is also important to note that DN* is much more computationally efficient with only an overhead of subtracting the mean while TPT requires performing gradient steps on 64 different augmented view of each test image.

Notably, TCL saw the largest gain of 7% on average in top-1 accuracy. On ImageNet1K particularly, TCL's top-1 accuracy is improved from 20.0% to 30.5% by adding DN. We believe the fact that TCL and ALBEF are pre-trained on a smaller dataset (4M & 14M images) than CLIP (400M images) resulted in TCL and ALBEF getting a larger improvement than CLIP, as DN may have helped ease the instability when the model is not adequately trained. All these improvements are achieved with a single layer that incurs minimal computational overhead. Lastly, comparing DN and DN* for TCL and ALBEF, we found that DN consistently gives a greater improvement than DN*, potentially because averaging with vanilla dot product halves the benefits from DN when the base models are instable due to inadequate training.

| | ImageNet1K | | Cifar100 | | SUN397 | | Stanford Cars | | Caltech 101 | | Flowers 102 | |
| | Acc@1 | Acc@5 | Acc@1 | Acc@5 | Acc@1 | Acc@5 | Acc@1 | Acc@5 | Acc@1 | Acc@5 | Acc@1 | Acc@5 |
|---|---|---|---|---|---|---|---|---|---|---|---|---|
| CLIP [50] | 61.0 | 87.4 | 63.9 | 88.7 | 56.1 | 89.4 | 58.6 | 90.9 | 82.3 | 95.0 | 62.1 | 83.8 |
| CALIP [14] | 61.2±0.2 | 87.5±0.1 | 64.2±0.1 | 88.9±0.0 | 56.1±0.1 | 89.3±0.1 | 58.7±0.0 | 90.1±0.0 | 82.5±0.1 | 95.1±0.0 | 62.2±0.1 | 83.4±0.0 |
| TPT [56] | **63.5** | 87.1 | 65.2 | 88.1 | **59.4** | 88.8 | **61.5** | 90.2 | 83.2 | 96.0 | **64.5** | 81.3 |
| CLIP + TTA | 62.4 | 88.5 | 66.0 | 90.5 | 56.9 | 90.0 | 60.8 | **92.3** | 82.5 | 95.4 | 62.5 | 84.0 |
| CLIP + TTA + DN | 63.0±0.0 | 88.8±0.0 | **67.5**±0.1 | **90.7**±0.1 | 58.8±0.2 | **91.0**±0.0 | 60.3±0.2 | 91.5±0.1 | **83.3**±0.1 | 95.4±0.0 | 63.1±0.1 | 84.3±0.1 |
| CLIP + TTA + DN * | 63.2±0.0 | **88.9**±0.0 | 67.1±0.1 | **90.7**±0.0 | 58.1±0.1 | 90.7±0.0 | **61.5**±0.1 | 92.2±0.0 | 83.1±0.0 | **95.5**±0.0 | 63.5±0.1 | **84.5**±0.0 |
| CLIP + DN | 61.6±0.1 | 87.7±0.1 | 65.5±0.2 | 89.2±0.2 | 57.9±0.1 | 90.5±0.1 | 57.5±0.1 | 90.5±0.1 | 82.9±0.1 | 94.9±0.1 | 62.7±0.1 | 84.1±0.1 |
| CLIP + DN* | 61.7±0.1 | 87.8±0.0 | 65.1±0.1 | 89.4±0.0 | 57.3±0.0 | 90.2±0.1 | 58.6±0.1 | 90.7±0.0 | 82.8±0.0 | 95.1±0.0 | 62.9±0.1 | 84.3±0.1 |
| TCL [68] | 20.0 | 42.9 | 39.1 | 68.2 | 28.6 | 63.4 | 2.0 | 8.7 | 58.8 | 80.1 | 24.4 | 42.6 |
| TCL + DN | **30.5**±0.2 | **54.3**±0.2 | **45.4**±0.1 | **73.1**±0.1 | **36.2**±0.2 | **70.8**±0.3 | **2.6**±0.1 | **10.7**±0.1 | **66.7**±0.1 | **81.8**±0.1 | **28.5**±0.1 | **48.4**±0.2 |
| TCL + DN* | 25.1±0.1 | 48.9±0.2 | 42.2±0.0 | 71.1±0.1 | 31.8±0.1 | 66.8±0.2 | 2.4±0.0 | 9.5±0.1 | 63.9±0.1 | 81.1±0.0 | 27.2±0.1 | 45.7±0.1 |
| ALBEF [33] | 37.3 | 65.8 | 38.5 | 65.7 | 45.6 | 81.5 | 25.0 | 61.0 | 66.0 | 86.3 | 26.9 | 49.1 |
| ALBEF + DN | **39.9**±0.2 | **68.5**±0.2 | **50.2**±0.1 | **77.0**±0.2 | **46.7**±0.1 | **82.4**±0.1 | **26.0**±0.2 | **62.0**±0.3 | **71.5**±0.3 | **88.9**±0.1 | **33.1**±0.2 | **54.4**±0.1 |
| ALBEF + DN* | 38.7±0.1 | 67.3±0.0 | 44.8±0.2 | 72.0±0.2 | 46.3±0.1 | 82.0±0.0 | **26.0**±0.1 | 61.7±0.2 | 68.9±0.4 | 87.6±0.3 | 29.9±0.1 | 51.9±0.1 |

Table 2: Zero-shot classification performance on ImageNet1K, Cifar100, SUN397, Stanford Cars, Caltech 101, and Flowers 102. Means for DN are estimated using 100 random unlabeled validation samples. Average accuracies and standard deviations are calculated with 5 random seeds.

## 4.5 Image Captioning Metric Results

Inspired by the success of CLIPScore [17] as a metric for image captioning, we conduct experiments to test whether DN and DN* can help various cross-modal representation models align better with human judgments. To compare the performance of different methods, we report the Kendall's $\tau_b$ or $\tau_c$ coefficients against human judgments on Flickr8k-expert, Flickr8k-cf [18] and THumb [24] in Table 3, following the standard practice in [17]. We also report the resulting accuracy for caption-caption preferences on Pascal-50S [51] in Table 7. Details about these datasets are given in appendix C.3. In the reference-based setting, for CLIP-ref, we follow [17] in using an averaged score given by:

$$\text{CLIP-ref}(x_0, y_0) = \text{H-mean}(\phi(x_0)^\intercal \psi(y_0), \max_{c \in R} \psi(c)^\intercal \psi(y_0)),$$

where H-mean is the harmonic mean and $R$ is the set of provided references. On the other hand, for a distribution normalized CLIP + DN-ref, we use an algebraic mean for simplicity:

$$\text{CLIP} + \text{DN-ref}(x_0, y_0) = \text{A-mean}(S_{(1)}(x_0, y_0), \max_{c \in R}(\psi(c) - \mu_y)^\mathsf{T}(\psi(y_0) - \mu_y)),$$

where A-mean is the algorithmic mean, $R$ is the set of provided references, $\mu_y$ is the mean of text representations, and $S$ is defined in Eqn. 9.

In Table 3, we report the results on Flickr8K-Expert, Flicker8K-CF, and THumb for all three base models. Some methods, such as the BLEU score, are not applicable in the reference-free setting and are thus only compared with in the reference-based setting. The results strongly support that adding DN significantly improves the $\tau$ correlation for all three base models on all dataset in the reference-free setting. In particular, CLIP + DN improves CLIP by 2.9% on Flickr8k-expert, 1.1% on Flickr8k-cf and 3.6 % on THumb, and CLIP + DN* performs comparably. In the reference-based setting, to give a better context, we also provide the performance of other reference-based SOTA metrics that are not contrastively trained for comparison. We find that CLIP and CLIP-ref outperform all of them. However, by simply adding a DN layer, CLIP-ref + DN again yields an average 1.3% gain across the board over previous leaders. Fur-

|  |  | Flickr8k-expert | Flickr8k-cf | THumb |
|---|---|---|---|---|
|  |  | $\tau_c$ | $\tau_b$ | $\tau_c$ |
| Ref-free | CLIP [17] | 51.4 | 34.3 | 19.9 |
|  | CLIP + TTA | 51.9 | 34.7 | 20.7 |
|  | CLIP + TTA + DN | 53.6 | **35.7** | 23.7 |
|  | CLIP + TTA + DN* | 53.5 | 35.5 | 22.7 |
|  | CLIP + DN | **54.3** | 35.4 | **23.5** |
|  | CLIP + DN* | 53.2 | 35.1 | 22.2 |
|  | TCL [68] | 31.0 | 20.6 | 8.1 |
|  | TCL + DN | **42.0** | **26.4** | **14.4** |
|  | TCL + DN* | 36.0 | 23.3 | 11.1 |
|  | ALBEF [33] | 24.9 | 15.4 | 0.9 |
|  | ALBEF + DN | **34.8** | **21.8** | **5.5** |
|  | ALBEF + DN* | 29.2 | 18.1 | 2.5 |
| Ref-based | BLEU-1 | 32.3 | 17.9 | 11.1 |
|  | BLEU-4 | 30.8 | 16.9 | 6.9 |
|  | CIDEr [63] | 43.9 | 24.6 | 13.8 |
|  | NUBIA* [23] | 49.5 | - | - |
|  | ViLBERTScore-F [69] | 50.1 | - | - |
|  | CLIP-ref [17] | 53.0 | 36.4 | 24.7 |
|  | CLIP + DN-ref | **55.3** | **37.0** | 25.8 |
|  | CLIP + DN*-ref | 54.3 | 36.9 | **26.2** |

Table 3: Image captioning metric results on Flickr8k-Expert, Flickr8k-CF, and THumb.

thermore, analogous to the results in cross-modal retrieval and zeroshot classification, we found that TTA is effective in improving on results of image captioning metric (CLIP + TTA Has on average 0.6% improvement over CLIP) but TTA combined with DN gives another 1.9% boost on average. A similar pattern is also observed in our experiment results on Pascal-50S in Appendix 7. Our main results on cross-modal retrieval, zero-shot classification, and image caption metrics provide extensive evidence that our proposed DN is a widely applicable modification that can improve the performance of cross-modal representation models across a variety of downstream applications.

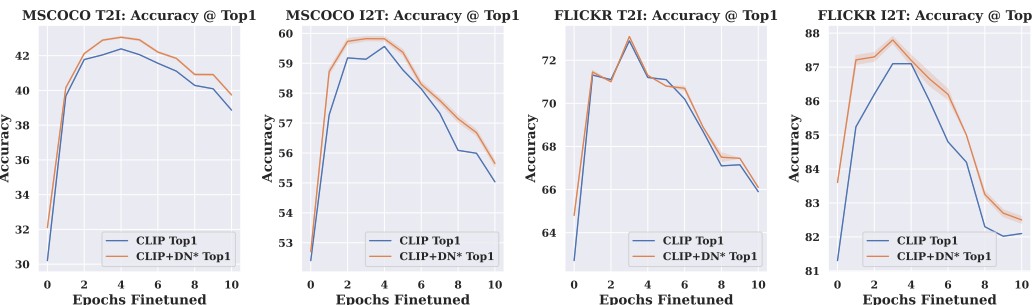

Figure 2: Comparison of the effects of fine-tuning between CLIP and CLIP + DN* for Acc@1. Results on MSCOCO's 5k test set are shown to the left, and Flickr30K's 1k test set are shown to the right. For each set, image-to-text retrieval (left) and image-to-text retrieval (right) are reported. For each of the 5 checkpoints we trained, we find its average accuracy and standard deviation with another 5 iterations random sampling for mean estimation, and plot the mean of these 5 accuracies and standard deviations from 5 independently fine-tuned checkpoints.

## 4.6 Ablation Study

### 4.6.1 Results on Different Variants of CLIP

In addition to the CLIP-B32 model used in most experiments, in Table 4 and Table 5, we investigate into the performance of DN* on other popular variants of CLIP models, including CLIPB16, CLIPL14

[50] that come with more parameters and smaller patch size, and CLIPB32-Laion [54] that is trained on a larger image-caption pair dataset. The results show that DN* brings a similar performance gain for CLIP with more parameters or trained on a larger dataset, implying the universal applicability of our proposed method irrespective to the size of the CLIP model.

| | MSCOCO (5K test set) | | | | | | Flickr30K (1K test set) | | | | | |
| | Image → Text | | | Text → Image | | | Image → Text | | | Text → Image | | |
| | R@1 | R@5 | R@10 | R@1 | R@5 | R@10 | R@1 | R@5 | R@10 | R@1 | R@5 | R@10 |
|---|---|---|---|---|---|---|---|---|---|---|---|---|
| CLIP(B16) + TTA | 53.6 | 77.5 | 85.1 | 33.8 | 58.7 | 69.1 | 85.4 | 97.9 | 99.1 | 66.6 | 89.0 | 93.7 |
| CLIP(B16) + TTA + DN* | **54.6** | **78.5** | **86.1** | **35.7** | **60.7** | **70.8** | **87.3** | **98.0** | **99.6** | **69.3** | **90.2** | **94.6** |
| CLIP(L14) + TTA | 57.7 | 80.1 | 87.8 | 36.8 | 61.3 | 71.2 | 88.4 | **98.9** | **99.9** | 69.9 | 90.6 | 94.8 |
| CLIP(L14) + TTA + DN* | **58.8** | **81.3** | **88.4** | **38.6** | **63.1** | **72.9** | **89.3** | 98.8 | 99.8 | **72.1** | **91.7** | **95.5** |
| CLIP(B32-Laion) + TTA | 58.5 | 80.9 | 88.1 | 40.0 | 65.8 | 76.0 | 85.8 | 96.7 | 98.9 | 71.1 | 91.4 | **94.8** |
| CLIP(B32-Laion) + TTA + DN* | **60.7** | **82.3** | **88.9** | **40.8** | **66.5** | **76.4** | **86.6** | **97.1** | 98.9 | **71.7** | **91.6** | 94.7 |

Table 4: Cross-modal retrieval performance on MSCOCO and Flickr30K in the zero-shot setting for more CLIP variants. Means for DN* are estimated using 100 random unlabeled validation samples. Average recalls are calculated with 5 random seeds.

| | ImageNet1K | | Cifar100 | | SUN397 | | Stanford Cars | | Caltech 101 | | Flowers 102 | |
| | Acc@1 | Acc@5 | Acc@1 | Acc@5 | Acc@1 | Acc@5 | Acc@1 | Acc@5 | Acc@1 | Acc@5 | Acc@1 | Acc@5 |
|---|---|---|---|---|---|---|---|---|---|---|---|---|
| CLIP(B16) + TTA | 67.1 | 91.5 | 67.7 | 90.1 | 60.0 | 91.4 | 63.3 | 93.1 | 84.5 | **96.7** | 69.8 | 84.8 |
| CLIP(B16) + TTA + DN * | **67.8** | **92.0** | **71.1** | **92.2** | **61.9** | **92.4** | **64.6** | **93.9** | **84.9** | 96.6 | **70.0** | **85.0** |
| CLIP(L14) + TTA | 73.1 | 93.4 | 77.6 | 94.0 | 62.1 | 92.5 | 76.3 | 97.7 | 86.0 | 97.3 | 72.8 | 89.1 |
| CLIP(L14) + TTA + DN * | **74.2** | **94.1** | **80.4** | **95.4** | **63.8** | **93.3** | **77.2** | **97.8** | **86.2** | 97.1 | **74.0** | 89.1 |
| CLIP(B32-Laion) + TTA | 66.9 | 89.4 | 75.7 | 94.0 | 63.9 | 93.5 | 87.1 | **99.2** | **87.3** | **97.7** | 70.3 | **86.4** |
| CLIP(B32-Laion) + TTA + DN * | **67.2** | **90.3** | **76.2** | **94.3** | **64.3** | **93.7** | 87.1 | 99.1 | 86.8 | 97.2 | **71.1** | 85.8 |

Table 5: Zero-shot classification performance on ImageNet1K, Cifar100, SUN397, Stanford Cars, Caltech 101, and Flowers 102. Means for DN* are estimated using 100 random unlabeled validation samples. Average accuracies are calculated with 5 random seeds.

### 4.6.2 Effect of Fine-Tuning

We also perform ablations on the effects of DN* on fine-tuned models. We fine-tune CLIP on the MSCOCO training set, for a total of 10 epochs on 4 Nvidia 2080Ti. We use the Adam optimizer with a learning rate of 1e-5 and a weight decay of 0.1. We then evaluated each checkpoint on the same MSCOCO test split used in Table 1, and estimate the mean for CLIP + DN* using 100 randomly sampled validation points. A total of 5 checkpoints were trained, and we evaluated each checkpoint for 5 iterations. For each checkpoint, we calculate the average accuracy and standard deviation, then plot the mean of the accuracies and standard deviations over all 5 checkpoints. We repeat this process for Flickr30K, using the same number of epochs, checkpoints, and model of GPUs. We again use the Adam optimizer with a learning rate of 1e-5, but with a weight decay of 0.02. We record accuracy scores at 1, 5, and 10 for both image-to-text retrieval and text-to-image retrieval on the two datasets. We present Acc@1 here, and leave Acc@5 and Acc@10 in Apendix D.1.

The following analysis focuses specifically on the case of Acc@1 for MSCOCO, but the general trend is also present in Flickr30K. As seen in Figure 2, fine-tuning allows for an improvement in accuracy over the zero-shot setting initially by 8.06% and 9.47% for CLIP + DN* and CLIP respectively in the text-to-image case, and 6.02% and 4.88% in the image-to-text case in the first epochs. But the fine-tuning performance gradually decays after that due to overfitting to the training set. At their highest peak, CLIP + DN* achieves 43.06% accuracy compared to CLIP's 42.39% for text-to-image retrieval, and 59.82% compared to 59.57% for image-to-text retrieval. We observe for both image-to-text retrieval and text-to-image retrieval that CLIP + DN* performs better than CLIP across all fine-tuning epochs by an average of 0.80% for text-to-image, and 0.60% for image-to-text case. A similar pattern is also observed in Flickr30K. Our experiments on the fine-tuning setting provide an affirmative answer to the last question we seek to answer, that DN* provides a non-trivial performance boost to both pre-trained and fine-tuned cross-modal representation models.

### 4.6.3 Number of Samples for Mean Estimation

We conduct experiments to examine the performance of DN* when different numbers of samples are used to estimate the mean. Using the ImageNet1K, Cifar100, SUN397, and Stanford Cars datasets,

we first test the retrieval accuracy of DN* by estimating the mean with the entire test set. We use the performance achieved here as an upper bound, because the mean is most reliable if it is directly calculated on all of the unlabeled testing samples. We compare it with the estimated means using 1, 10, and 100 random samples from the training set, and present our results in Table 6.

As shown in the table, decreasing the number of samples to just 100 from the training set results in an accuracy drop of no more than 0.2% compared to the upper bound, with an average of only 0.09%. We find that even decreasing the number of samples to as few as 10 does not significantly hurt the result. For 10 samples, there is an average drop in accuracy of 0.14%, with the greatest difference of no more than 0.4%. Moreover, for CLIP, TCL, and ALBEF, we find that estimating the mean with as few as 10 samples still improved accuracy over the base model by an average of 1.96% for Acc@1, and 1.92% for Acc@5. Furthermore, our results are generally stable despite the low number of sample points used to estimate the mean. Across 5 random samplings, the standard deviation of our accuracy is on average 0.08% for 100 training set samples, and 0.22% for 10 training set samples. Even in the extreme case of just 1 sample, we still observe a slight improvement of 1.15% over the base model averaged across Acc@1 and Acc@5, but with a slightly larger standard deviation of 0.77%. These ablation results show that DN* can generalize very well even in the setting of extremely scarce data, possibly because DN* only needs to know the global mean of the distribution and does not require fine-grained local details.

| | ImageNet1K | | Cifar100 | | SUN397 | | Stanford Cars | |
|---|---|---|---|---|---|---|---|---|
| | Acc@1 | Acc@5 | Acc@1 | Acc@5 | Acc@1 | Acc@5 | Acc@1 | Acc@5 |
| CLIP [50] | 61.0 | 87.4 | 63.9 | 88.7 | 56.1 | 89.4 | 58.6 | 90.9 |
| CLIP + DN* (test) | 61.8 | 87.9 | 65.1 | 89.4 | 57.3 | 90.1 | 58.7 | 90.7 |
| CLIP + DN* (100) | $61.7 \pm 0.0$ | $87.8 \pm 0.0$ | $65.1 \pm 0.1$ | $89.4 \pm 0.0$ | $57.3 \pm 0.0$ | $90.2 \pm 0.1$ | $58.6 \pm 0.1$ | $90.7 \pm 0.0$ |
| CLIP + DN* (10) | $61.6 \pm 0.1$ | $87.8 \pm 0.1$ | $65.1 \pm 0.2$ | $89.2 \pm 0.1$ | $57.1 \pm 0.1$ | $90.1 \pm 0.1$ | $58.3 \pm 0.1$ | $90.7 \pm 0.1$ |
| CLIP + DN* (1) | $60.6 \pm 0.4$ | $87.2 \pm 0.2$ | $64.1 \pm 0.6$ | $88.4 \pm 0.6$ | $56.9 \pm 0.5$ | $89.7 \pm 0.2$ | $56.5 \pm 0.4$ | $89.7 \pm 0.5$ |
| TCL [68] | 20.0 | 42.9 | 39.1 | 68.2 | 28.6 | 63.4 | 2.0 | 8.7 |
| TCL + DN* (test) | 25.4 | 49.4 | 42.2 | 71.1 | 31.9 | 66.9 | 2.4 | 9.5 |
| TCL + DN* (100) | $25.1 \pm 0.1$ | $48.9 \pm 0.2$ | $42.2 \pm 0.0$ | $71.1 \pm 0.1$ | $31.8 \pm 0.1$ | $66.8 \pm 0.2$ | $2.4 \pm 0.0$ | $9.5 \pm 0.1$ |
| TCL + DN* (10) | $25.3 \pm 0.1$ | $49.1 \pm 0.2$ | $42.2 \pm 0.2$ | $70.9 \pm 0.1$ | $31.9 \pm 0.4$ | $66.8 \pm 0.5$ | $2.3 \pm 0.1$ | $9.5 \pm 0.0$ |
| TCL + DN* (1) | $25.5 \pm 0.3$ | $49.2 \pm 0.3$ | $40.8 \pm 0.7$ | $69.4 \pm 0.9$ | $31.4 \pm 0.5$ | $65.8 \pm 0.9$ | $2.3 \pm 0.1$ | $9.5 \pm 0.3$ |
| ALBEF [33] | 37.3 | 65.8 | 38.5 | 65.7 | 45.6 | 81.5 | 25.0 | 61.0 |
| ALBEF + DN* (test) | 38.7 | 67.4 | 45.0 | 72.1 | 46.2 | 81.9 | 26.0 | 61.8 |
| ALBEF + DN* (100) | $38.7 \pm 0.1$ | $67.3 \pm 0.0$ | $44.8 \pm 0.2$ | $72.0 \pm 0.2$ | $46.3 \pm 0.1$ | $82.0 \pm 0.0$ | $26.0 \pm 0.1$ | $61.7 \pm 0.2$ |
| ALBEF + DN* (10) | $38.7 \pm 0.1$ | $67.3 \pm 0.1$ | $45.1 \pm 0.6$ | $72.2 \pm 0.5$ | $45.9 \pm 0.3$ | $81.7 \pm 0.2$ | $25.8 \pm 0.3$ | $61.3 \pm 0.6$ |
| ALBEF + DN* (1) | $37.3 \pm 1.2$ | $65.6 \pm 1.4$ | $44.3 \pm 2.8$ | $71.1 \pm 3.1$ | $45.4 \pm 0.3$ | $81.4 \pm 0.6$ | $25.0 \pm 0.6$ | $60.1 \pm 1.2$ |

Table 6: Ablation study on the effect of the number of samples to estimate the mean for ImageNet1K, Cifar100, SUN397 and Stanford Cars. Parenthesized is the number of unlabeled samples from the training set. (test) means that DN is conducted on the entire unlabeled test set. Mean and standard deviation are calculated based on 5 random samplings.

# 5   Limitations and Social Impact

The success of distribution normalization builds on access to a small amount (10-100) of unlabeled data from the test distribution. Although much cheaper and more easily accessible than the setting of fine-tuning or few-shot learning where more labeled data is needed, the use of unlabeled data can still be a potential limitation in cases when the test distribution is entirely unknown. This work builds on top of a recent line of work that consolidates vast amounts of data from public resources on the Internet to learn a joint representation space of images and text. Therefore, it is possible that DN might inherit or strengthen the social biases in those large visual-language models.

# 6   Conclusion

In this paper, we identify a mismatch between the pre-training objective of cross-modal representation models like CLIP and its downstream use with a direct dot product. To address this problem, we propose Distribution Normalization (DN) that not only displays a significant advantage over direct dot product in a wide variety of settings but is also straightforward to implement. However, though being extremely sample efficient, DN still requires estimating a mean separately for each distribution and it can be of interest if a universal mean can be found so that DN is more easily applied to a wider range of downstream applications. Future research can also explore the implications of DN to the contrastive training process of cross-modal representation models.

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

## A  More Related Works

**Contrastive Learning** Contrastive learning has become a popular method for self-supervised representation learning, where the goal is to learn embedding functions such that semantically similar samples are closer in embeddings space while semantically dissimilar samples are embedded further apart [59, 66, 47, 70, 72, 13]. Among various contrastive learning methods, this paper specifically focuses on the InfoNCE loss, which optimizes the probability of correctly identifying the positive sample [62], as it has been one of the most frequently used objectives in recent contrastive learning tasks [16, 5, 3, 4].

**Image Caption Evaluation** Image caption evaluation is the task of automatically scoring the quality of a caption given the corresponding image and optionally reference captions. The quality of image caption metrics are evaluated with respect to its correlation with human ratings. While traditional image caption metrics such as BLEU-4 [45], ROUGE-L [38], METEOR [2], CIDEr [63], and SPICE [1] evaluate the candidate caption by measuring the n-gram overlap of the candidate with the references, more contemporary neural metrics [22, 69, 23, 30, 8] judge the candidate caption by comparing the neural embeddings of the caption with that of the image and references, and are much more flexible. Among them, the one most related to our work is CLIPScore [17]. By simply passing test-time image and caption pairs through the pre-trained CLIP image and text encoders and taking the dot product, CLIPScore achieves state-of-the-art performance using only the candidate caption and the corresponding image without the need for references.

## B  Base Representation Model Details

This paper examines the effect of a Distribution Normalization layer on the following state-of-the-art open-sourced cross-modal representation models:

**CLIP** [50] is one of the earliest pioneers that successfully created a joint representation space of images and text through large-scale contrastive learning. It is trained on over 400M image-caption pairs collected by the authors. They have open-sourced multiple versions of pretrained models and this paper examines the commonly used version "ViT-B/32".

**ALBEF** [33] is one of the later variants of CLIP that adds a multi-modal encoder to capture the interplay between images and text by predicting if they are paired samples or hard negatives. Additional momentum distillation loss and masked language modeling loss are also added to improve the model performance. Two versions of pre-trained models trained on dataset consisting of 4M and 14M unique images are released and this work uses the latter. To focus on the effect of DN on cross-modal representations, we only keep the image and text encoder component of ALBEF and do not use the multi-modal encoder component.

**TCL** [68] is another state-of-the-art CLIP variant. It inherits the same model architecture as ALBEF, but adds triplet contrastive loss including Cross-modal Alignment, Intra-modal Contrastive, and Local MI Maximization. Their model is trained on 4M unique images. As with ALBEF, we only keep the image and the text encoder component.

## C  Datasets Details

There is a total of 12 datasets mentioned in this paper, as described below.

### C.1  Cross-modal Retrieval Datasets

**MSCOCO** [39] is a multi-purpose dataset known for its rich compatibility with object detection, segmentation, and image captioning tasks. It contains 118K images in its training split and more than 5K images in its test split.

**Flicker30K** [48] is a popular benchmark for sentence-based picture portrayal. It contains 31K images with each image having five reference sentences generated by human annotators. We took a train-test split following [68] and [33], which involves a selection of 30K images for fine-tuning and 1K images for testing.

## C.2 Zeroshot Classification Datasets

**ImageNet1K** [9] is the most well-known dataset for image classification that contains more than 1200K images in its training set and 100K images in its test set and covers 1000 categories of objects. Accuracy on ImageNet1K is especially widely used for evaluating state-of-the-art image classification models.

**Cifar100** [28] is a dataset containing 100 classes of objects, with each class having 500 training images and 100 test images.

**SUN397** [67], the Scene UNderstanding database, contains 899 categories and about 130K images. 397 well-sampled categories are available for benchmarking state-of-the-art algorithms for scene recognition tasks.

**Stanford Cars** [27] is an image classification dataset dedicated for cars. It collects about 16K images for 196 types of cars and adopts a 50-50 train-test split. Results on Stanford Cars test our method's effectiveness in improving special-purpose classification models.

**Caltech101** [31] is a dataset for object recognition tasks. It contains 101 object categories, with varying numbers of images in each category (between 40 to 800 images per category). The dataset has over 9,000 images in total collected from the internet. It is a popular benchmark for evaluating the performance of algorithms on general-purpose single-object recognition tasks.

**Flowers102** [44] is an image classification dataset focused on flowers, comprising 102 distinct categories of flowers common to the UK. It includes a diverse set of images, with each class containing between 40 to 258 images. The dataset comprises over 8,000 images.

## C.3 Image Captions Evaluation Datasets

**Flickr8K-Expert** [18] is a curated subset of the Flickr8K dataset that contains 17K human ratings of image-caption pairs. Each human score corresponds to one pair and is from 1 to 4 (1 means the caption is irrelevant, 4 means the caption describes the image fully correctly.)

**Flicker8K-CF** [18] is a similar dataset gathered from CloudFlower that contains 48K image-caption pairs and 145K binary ratings of these pairs.

**THumb** [24] is a dataset containing some machine- and human-generated captions from the MSCOCO dataset. It has 500 images, each with 5 candidates captions that are evaluated by human annotators.

**Pascal-50S** [51] contains 4K caption-caption pairs with each pair describing the same image. For all pairs, annotators give a preference on which caption in the pair provides better description of the image. Caption-caption pairs are categorized based on the origins of the captions (see Table 7 for details about categories.)

# D   Additional Results

## D.1   Fine-tuning Epochs on MSCOCO and Flickr30K

In Figure 3, we present the remaining two plots of fine-tuning ablations for MSCOCO. CLIP + DN* does better than CLIP by an average of 0.52% for Acc@5, and 0.37% for Acc@10. Both graphs observe a peak in accuracy around 4-5 epochs. For Acc@5, we observe an initial improvement of 9.10% for CLIP + DN* after just 1 epoch, and 11.10% for CLIP on text-to-image retrieval. These respective numbers are 5.17% and 4.39% for image-to-text retrieval. For Acc@10, we observe an initial improvement of 8.53% for CLIP + DN* and 10.03% for CLIP on text-to-image retrieval, and 3.14% and 3.14% respectively for image-to-text retrieval.

In Figure 4, we present the effects of finetuning on Acc@5 and Acc@10 for Flickr30K. Again, CLIP+DN* does better than CLIP by an average of 0.22% for Acc@5 and 0.14% for Acc@10. All results yield a similar trend in support of the conclusion we have made in Section 4.6.2.

|          | HC | HI | HM | MM | Mean |
|----------|------|------|------|------|------|
| CLIP [17] | 55.0 | 99.2 | 96.9 | 71.8 | 80.7 |
| CLIP + TTA | 55.2 | 99.2 | 96.8 | 71.8 | 80.8 |
| CLIP + TTA + DN | 55.1 | 99.2 | **97.4** | 73.7 | 81.4 |
| CLIP + TTA + DN* | 55.2 | 99.3 | **97.4** | 72.9 | 81.2 |
| CLIP + DN | **56.1** | 99.1 | 97.3 | **73.9** | **81.6** |
| CLIP + DN* | 55.6 | **99.3** | 97.3 | 73.6 | 81.4 |
| TCL [68] | 52.4 | 91.8 | 54.7 | 63.4 | 65.6 |
| TCL + DN | 52.4 | **96.7** | **65.7** | **66.4** | **70.3** |
| TCL + DN * | **52.7** | 93.8 | 59.0 | 64.3 | 67.4 |
| ALBEF [33] | 56.9 | 98.1 | **82.5** | 67.2 | **76.2** |
| ALBEF + DN | 56.1 | 97.9 | 81.2 | 67.0 | 75.6 |
| ALBEF + DN* | 56.9 | 98.1 | 81.7 | **67.8** | 76.1 |
| BLEU-4 | 60.4 | 90.6 | 84.9 | 54.7 | 72.6 |
| CIDEr [63] | 65.1 | 98.1 | 90.5 | 64.8 | 79.6 |
| ViLBERTScore-F [69] | 49.9 | 99.6 | 93.1 | 75.8 | 79.6 |
| CLIP -ref [17] | **62.4** | 99.7 | 96.7 | 73.0 | 83.0 |
| CLIP + DN -ref | 60.8 | 99.6 | **97.8** | 75.1 | **83.3** |
| CLIP + DN* -ref | 61.1 | 99.7 | 97.3 | 74.8 | 83.2 |

Table 7: Accuracy results on Pascal-50S given different categories of caption-caption pairs. HC means two correct human-generated captions. HI means two human-generated captions with one incorrect. HM means a human-generated and a machine-generated caption. MM means two machine-generated captions.

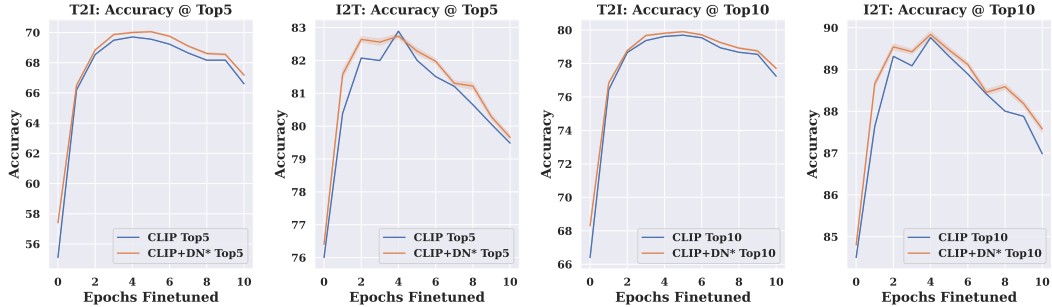

Figure 3: Comparison of the effects of fine-tuning between CLIP and CLIP + DN* on MSCOCO's 5k test set. We report text-to-image retrieval (left) and image-to-text retrieval (right) for Acc@5 and Acc@10. The average accuracy over 5 checkpoints trained with 5 random seeds is plotted. For each of the 5 checkpoints we trained, we find its average accuracy and standard deviation with another 5 iterations random sampling for mean estimation, and plot the mean of these 5 accuracies and standard deviations from 5 independently fine-tuned checkpoints.

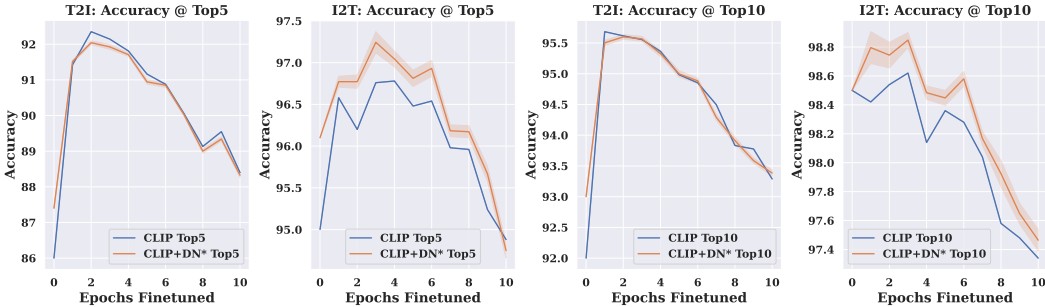

Figure 4: Comparison of the effects of fine-tuning between CLIP and CLIP + DN* on Flickr30K's 1k test set. We report text-to-image retrieval (left) and image-to-text retrieval (right) for all Acc@5 and Acc@10. The average accuracy over 5 checkpoints trained with 5 random seeds is plotted. For each of the 5 checkpoints we trained, we find its average accuracy and standard deviation with another 5 iterations random sampling for mean estimation, and plot the mean of these 5 accuracies and standard deviations from 5 independently fine-tuned checkpoints.

## D.2 Information Loss in Approximation

Without considering computational efficiency, for each test sample, we can iterate over the entire unlabeled reference set to calculate a similarity measure given in Eqn.5:

$$S_{full}(x_0, y_0) = \mathbb{E}_{y_1 \sim \mathcal{D}_T} e^{\phi(x_0)^\intercal [\psi(y_1) - \psi(y_0)]/\tau} + \mathbb{E}_{x_1 \sim \mathcal{D}_T} e^{[\phi(x_1) - \phi(x_0)]^\intercal \psi(y_0)/\tau}. \tag{10}$$

To investigate how much information might be lost from the first-order approximation in Section 3.2.2 and algebraic mean to geometric mean conversion in Section 3.3 that simplifies Eqn.10 to distribution normalization in Eqn.9, we carry out experiments to compare them back to back in the task of image captioning metric and present our results in Table 8 in terms of their correlation with human judgments on image captioning datasets. Surprisingly, in all the base models and datasets that we have studied, we did not notice any significant difference between DN and Eqn. 10 (difference $< 0.1\%$). This shows that higher-order information contributes negligibly to the downstream applications compared to only taking the mean of the distribution, and taking an algebraic mean achieves a similar effect as taking a geometric mean.As a conclusion, we found that DN provides equivalent performance to the original Eqn.5 without incurring expensive computational costs.

### D.2.1 Pixel-Level Normalization

A potential alternative to distribution normalization which normalizes the data on the representation level is a commonly used trick that normalizes the data (mostly images) on the input level (pixel level), which we refer as pixel-level normalization. In Table 9, we present our the comparison results between CLIP + pixel-norm and our proposed methods. However, although using the same data for normalization on the representation level as DN and DN* yields a large gain over vanilla CLIP, changing the normalization to the pixel level wipes out all the improvements. We hypothesize that this is because the difference of a constant vector (representation mean) vanishes while the input goes through a large neural network like CLIP.

|  | Flickr8k-expert | Flickr8k-cf | THumb |
|---|---|---|---|
|  | $\tau_c$ | $\tau_b$ | $\tau_c$ |
| CLIP [50, 17] | 51.4 | 34.3 | 19.9 |
| CLIP + DN | 54.3 | 35.4 | 23.5 |
| CLIP + DN (Eqn.10) | 54.3 | 35.4 | 23.4 |
| TCL [68] | 31.0 | 20.6 | 8.1 |
| TCL + DN | 42.0 | 26.4 | 14.4 |
| TCL + DN (Eqn.10) | 41.8 | 26.4 | 14.3 |
| ALBEF [33] | 24.9 | 15.4 | 0.9 |
| ALBEF + DN | 34.8 | 21.8 | 5.5 |
| ALBEF + DN (Eqn.10) | 34.8 | 21.8 | 5.5 |

Table 8: Abaltion study on comparison with distribution normalization and full Eqn.5 on Flickr8k-Expert, Flickr8k-CF, and THumb.

|  |  | Flickr8k-expert | Flickr8k-cf | THumb |
|---|---|---|---|---|
|  |  | $\tau_c$ | $\tau_b$ | $\tau_c$ |
| | CLIP [17] | 51.4 | 34.3 | 19.9 |
| Ref-free | CLIP + pixel-norm | 51.3 | 34.3 | 19.4 |
| | CLIP + DN | 54.3 | 35.4 | **23.5** |
| | CLIP + DN* | 53.2 | 35.1 | 22.2 |

Table 9: Ablation study on pixel-level normalization on Flickr8k-Expert, Flickr8k-CF, and THumb.

