# OpenReview forum: "Test-Time Distribution Normalization for Contrastively Learned Visual-language Models"
_NeurIPS.cc/2023/Conference — NeurIPS 2023 poster_

### Official Review · Reviewer_XtRE · 2023-06-25

**Soundness:** 3 good
**Presentation:** 4 excellent
**Contribution:** 3 good
**Rating:** 6
**Confidence:** 5

**Summary:**

This paper reveals a mismatch between the pre-training objective of contrastively trained vision-language models and their downstream usage. The authors propose  Distribution Normalization (DN) to solve this problem. The results on a wide variety of downstream tasks show the effectiveness of the proposed method.


**Strengths:**

1. The finding about the misalignment between pretraining and downstream tasks for contrastively trained vision-language models is important.

2. The proposed method, DN, is simple but effective, and also conveniently implemented in practice.

3. The experiments on various tasks show the effectiveness of the proposed method, and also there are sufficient ablation studies.


**Weaknesses:**

Overall, I think this paper is well-presented and reveal an important problem in contrastively trained vision-language models. However, I still have the below concerns:

1. A recent work, TPT [1], is missing, which aims to learn prompts for vision-language models at the test-time. I recommend that the authors will compare DN with TPT and integrate DN into the TPT’s framework.

2. I suggest that the authors will modify “visual-language” to “vision-language” in the next version.

3. Another concern is that because DN  needs a small amount of unlabeled data, its setting actually is not the zero-shot one, thus I suggest the authors will refine the related descriptions, e.g. Line 175.


[1] Test-Time Prompt Tuning for Zero-Shot Generalization in Vision-Language Models. NeurIPS 2023.


**Questions:**

Please see the weaknesses.

**Limitations:**

The authors have adequately discussed the limitations of this paper.

---

> ### Author Rebuttal · Authors · 2023-08-09
>
>  ### 1, Add TPT baseline
> Thanks for providing the related work, and we will include the comparisons in a revised version. Here are the results of TPT in our setting and a comparison of our results. (top1/top5). We found that our CLIP+TTA+DN* achieves a comparable top-1 accuracy and a higher top-5 accuracy compared to TPT, while at the same time being much more efficient as TPT needs to do gradient updates for each test sample.
> |  |imagenet1k |   Cifar100   |   SUN397     | Stanford Cars |  Caltech101   |   Flowers102     |avg |
> |-----------|---------------|------|--------|---------------|------|--------|---|
> | TPT     |  63.5/87.1       | 65.2/88.1 | 59.4/88.8   | 61.5/90.2        | 83.2/96.0 | 64.5/81.3   | 66.2/88.6 |
> | CLIP+TTA+DN*    |   63.2/88.9     | 67.1/90.7  |  58.1/90.7  |   61.5/92.2     | 83.1/95.5 |  63.5/84.4  | 66.1/90.4 |
>
> ### 2, Change to vision-language models and refine related descriptions
> Thanks for the reminder. We will change to “vision-language models” and refine the related descriptions to include the need for extra unlabeled samples.

---

### Official Review · Reviewer_5JmX · 2023-06-30

**Soundness:** 3 good
**Presentation:** 2 fair
**Contribution:** 2 fair
**Rating:** 6
**Confidence:** 4

**Summary:**

This paper proposes distribution normalization (DN) for contrastively trained vision-language models. The idea is motivated by an analysis of the InfoNCE loss. The authors identify that the common practice of taking dot product for zero-shot inference is only a zero-order approximation of the InfoNCE loss. The proposed method improves the dot product inference by using unlabeled test data to make a first-order approximation.

**Strengths:**

1. The analysis in Section 3.2 is intuitive and motivates the proposed method.
2. The method is efficient: it only needs to subtract the mean of the text/image features of the test data.
3. The experiment section evaluates the method on multiple tasks and various contrastively pre-trained models.


**Weaknesses:**

1. There are some missing baselines for CLIP zero-shot classification. For example, [1] proposes test-time prompt tuning for CLIP, which is less efficient than DN, but it works on a single test sample. [2] uses multiple test samples from the same distribution to do logit normalization, which is similar to the proposed method.


[1]. Shu, Manli, et al. "Test-time prompt tuning for zero-shot generalization in vision-language models.", NeurIPS 2022.
[2]. Allingham, James Urquhart, et al. "A Simple Zero-shot Prompt Weighting Technique to Improve Prompt Ensembling in Text-Image Models.", ICML 2023.

**Questions:**

1. Regarding performance, I find DN's improvement margins on CLIP to be smaller than on other models. Any idea what would cause this discrepancy between different contrastively-trained models?

**Limitations:**

1. As mentioned by the author, as a limitation of this work, DN requires access to multiple test samples and assumes they are all from the same distribution. Such an assumption may not hold in practical settings.
2. Many existing works have explored The idea of using test data for normalization at test time [2, 3, 4]. The authors should consider citing them and discussing their connections and differences. In addition, there is another work also analyzes the loss functions for test-time adaptation [5].

[3]. Schneider, Steffen, et al. "Improving robustness against common corruptions by covariate shift adaptation." NeurIPS 2020.
[4]. Wang, Dequan, et al. "Tent: Fully test-time adaptation by entropy minimization.", ICLR 2021.
[5]. Goyal, Sachin, et al. "Test-time adaptation via conjugate pseudo-labels.", NeurIPS 2022.

---

> ### Author Rebuttal · Authors · 2023-08-09
>
> ### 1, Two more baselines missing
> Thanks for providing the related work, and we will include the comparisons in a revised version. Here are the results of TPT in our setting and a comparison to our results. (top1/top5). We found that our CLIP+TTA+DN* achieves a comparable top-1 accuracy and a higher top-5 accuracy compared to TPT, while at the same time being much more efficient as TPT needs to do gradient updates for each test sample.
> |  |imagenet1k |   Cifar100   |   SUN397     | Stanford Cars |  Caltech101   |   Flowers102     |avg |
> |-----------|---------------|------|--------|---------------|------|--------|---|
> | TPT     |  63.5/87.1       | 65.2/88.1 | 59.4/88.8   | 61.5/90.2        | 83.2/96.0 | 64.5/81.3   | 66.2/88.6 |
> | CLIP+TTA+DN*    |   63.2/88.9     | 67.1/90.7  |  58.1/90.7  |   61.5/92.2     | 83.1/95.5 |  63.5/84.4  | 66.1/90.4 |
>
> For Zero-shot Prompt Weighting (ZPW), we agree with the reviewer that this would be an interesting baseline to have, but we would also like to emphasize that our proposed method is, in principle, meant to be auxiliary to ZPW. Specifically, since ZPW only weighs the importance of the dot-product similarity of each prompt template from a prompt pool, DN can still be applied by subtracting the mean of these embeddings from the logits. This also does not interfere with the weighted average of logits cross-product with prompt scores. Further, results from the ZPW paper are actually tested on a slightly different dataset split and we have previously looked into the released code for this paper but found that it is currently still in development with several unresolved dependencies. Should the stable version of the code be released prior to submission, we would be happy to include the performance of ZPW + DN as an additional baseline.
>
> ### 2, Are Improvements on CLIP smaller than other methods?
> Yes, your observation is correct. We have provided a general response on this issue and kindly ask the reviewer to refer to the “Performance Gain” general response for this concern.
>
>
> ### 3, Differences with other related works.
> Thanks for providing these related works. [3] estimates statistics for batch normalization within the large models. This method is not applicable to CLIP because CLIP does not have batch normalization to avoid information about the batch leaked in the pre-training. [2] and [3] involves minimizing a loss over all test samples. They require more unlabeled samples, are less efficient than DN, and cannot handle general multimodal alignment tasks such as cross-modal retrieval and image caption metrics. We will add detailed comparisons with those works in a revised version.
>
> [3]. Schneider, Steffen, et al. "Improving robustness against common corruptions by covariate shift adaptation." NeurIPS 2020.
>
> [4]. Wang, Dequan, et al. "Tent: Fully test-time adaptation by entropy minimization.", ICLR 2021.
>
> [5]. Goyal, Sachin, et al. "Test-time adaptation via conjugate pseudo-labels.", NeurIPS 2022.

---

> > ### Comment · Reviewer_5JmX · 2023-08-13
> >
> > Thank the authors for their response and additional results. I have carefully read the author’s reply and other reviewers’ comments. My questions have been properly answered, and I find the additional evaluation (including those suggested by other reviewers) to further support the proposed method. Therefore, I’m updating my recommendation to weak accept.

---

> > > ### Author Response · Authors · 2023-08-13
> > >
> > > We would like to thank the reviewer for seeing the value of the work and appreciate the reviewer for raising the score!

---

### Official Review · Reviewer_UGnD · 2023-07-04

**Soundness:** 1 poor
**Presentation:** 3 good
**Contribution:** 2 fair
**Rating:** 6
**Confidence:** 5

**Summary:**

CLIP is trained using InfoNCE loss where positive and negative pair alignment is done during training. However, at inference, we simply take the dot product with text embeddings which is not the optimal/similar to the way pretraining was done. Authors propose to rectify this inference and training objective misalignment, based on a first order approximation of InfoNCE loss. Specifically, they subtract the mean representation (calculated over a validation set) from the test image and label/caption representation before applying the dot product. The empirical evaluations of the proposed approach however do not give strong enough gains, especially on CLIP.

**Strengths:**

- Approach is easy to implement.
- Paper writing is generally smooth and clear. However, section around Eqn 8 (L130 to L133) should be improved and is not clear.


**Weaknesses:**

- Any reason why the proposed approach does not work well on MSCOCO retrieval image to text.
- Empirical gains are pretty nominal, especially for the CLIP models. The authors average their gains over CLIP and other not widely used models (TCL and ALBEF). However, the gains seems to be much lower on CLIP itself.
- Moreover, as mentioned in L252, it seems that there are larger gains with models pretrained on smaller datasets. This suggests that as the pretraining corpus size is increased, say we use CLIP pretrained on LAION5B, there may not be any gains. Can the reviewers test their approach on CLIP pretrained on larger datasets like DataComp XL ( https://huggingface.co/laion/CLIP-ViT-L-14-DataComp.XL-s13B-b90K ) or LAION5B.
- Authors should test their approach on larger and more standard CLIP architecture like ViT-B-16 and ViT-L, which have higher zeroshot accuracies.
- Table 2, authors report 61.0 as the zeroshot accuracy of CLIP ViT-B-32, which is however 63.2 (https://arxiv.org/pdf/2103.00020.pdf). Can the authors please clarify this.
- Can the authors provide finetuning experiments on image classification datasets as well like ImageNet or StanfordCars/Flowers/etc. considered in Table 2.
- (typo) Eq 5, D_s should be D_t in expectation.
- The supplementary material is supposed to be submitted as a separate PDF and not part of the main paper, to the best of my knowledge.


**Questions:**

Please see the weakness section.

**Limitations:**

Yes

---

> ### Author Rebuttal · Authors · 2023-08-09
>
>
> ### 1, proposed approach does not work well on MSCOCO retrieval image to text.
> In the setting of mscoco image to text, while DN does not perform well with CLIP-B32, it has shown effectiveness with other CLIP variants such as CLIP-B16 and CLIP-L14 as follows. A method's performance may depend on the specific characteristics of the dataset and the pre-trained model, and it’s important to highlight that it has shown improvement in the vast majority of settings. This indicates that DN is generally a reliable enhancement for multimodal models.
>
>
> ### 2, Improvements not strong, especially on CLIP
> We have provided a general response on this issue and kindly ask the reviewer to refer to the “Performance Gain” general response for this concern.
>
> ### 3, Improvements on CLIP are smaller compared to ALBEF and TCL. What about CLIP-B16, CLIP-L, and CLIP pre-trained on DataComp XL?
> We have conducted experiments on CLIP-B16, CLIP-L14 and found a similar if not larger improvement of DN for those larger models compared to CLIP-B32 used in the paper. Although the improvement is smaller for CLIP pre-trained on laion for the classification task, its improvement on cross-modal retrieval is still significant and comparable to the improvement for the original CLIP. This shows that DN is most effective when the downstream task resembles the contrastive pre-training procedure as cross-modal retrieval datasets contain diverse image-caption pairs similar to the pre-training dataset, as the purpose of DN is to align the downstream use with the pre-training objective.
>
> Furthermore, we believe the large improvement of DN on ALBEF and TCL is due to the large distribution shift from the small pre-training dataset to other datasets. This shows that DN is particularly successful in adapting to out-of-distribution scenarios. This out-of-distribution adaptation ability can be important even for models pre-trained on large datasets, especially when applied to real-world sustained distributions different from the training set.
>
> Cross-modal Retrieval (Top1/Top5/Top10)
> |  | MSCOCO I2T|    MSCOCO T2I       | Flickr30k  I2T|      Flickr30k T2I   | avg |
> |-----------|---------------|------|--------|---------------|---|
> | CLIP-B16 + TTA |    53.6/77.5/85.1     | 33.8/58.7/69.1 |  85.4/97.9/99.1  |    66.6/89.0/93.7      | 59.9/80.8/86.8 |
> | CLIP-B16 + TTA + DN* |    54.6/78.5/86.1      |35.7/60.7/70.8  | 87.3/98.0/99.6  |     69.3/90.2/94.6      | 61.7/81.9/87.8 |
> | CLIP-L14 + TTA |    57.7/80.1/87.8      | 36.8/61.3/71.2 | 88.4/98.9/99.9   |   69.9/90.6/94.8       | 63.2/82.7/88.4 |
> | CLIP-L14 + TTA + DN* |   58.8/81.3/88.4       | 38.6/63.1/72.9 | 89.3/98.8/99.8  |   72.1/91.7/95.5        | 64.7/83.7/89.2 |
> | CLIP-B32-Laion + TTA |     58.5/80.9/88.1     | 40.0/65.8/76.0 | 85.8/96.7/98.9   |   71.1/91.4/94.8       | 63.9/83.7/89.5 |
> | CLIP-B32-Laion + TTA + DN* |    60.7/82.3/88.9      | 40.8/66.5/76.4 | 86.6/97.1/98.9  |   71.7/91.6/94.7        | 65.0/84.4/89.7 |
>
>
> Classification (Top1/Top5)
> |  |imagenet1k |   Cifar100   |   SUN397     | Stanford Cars |  Caltech101   |   Flowers102     | avg |
> |-----------|---------------|------|--------|---------------|------|--------|---|
> | CLIP-B16 + TTA     | 67.1/91.5    | 67.7/90.1 | 60.0/91.4  |  63.3/93.1      | 84.5/96.7 | 69.8/84.8   | 68.7/91.3 |
> | CLIP-B16 + TTA + DN*    |  67.8/92.0   | 71.1/92.2 | 61.9/92.4  |  64.6/93.9      | 84.9/96.6  |  70.0/85.0  | 70.1/92.0 |
> | CLIP-L14 + TTA     |  73.1/93.4   | 77.6/94.0 | 62.1/92.5  |   76.3/97.7     | 86.0/97.3 |  72.8/89.1  | 74.6/94.0 |
> | CLIP-L14 + TTA  + DN*   |  74.2/94.1   | 80.4/95.4 | 63.8/93.3  |  77.2/97.8      | 86.2/97.1 | 74.0/89.1   | 76.0/94.5  |
> | CLIP-B32-Laion + TTA     | 66.9/89.4    | 75.7/94.0 | 63.9/93.5  |   87.1/99.2     | 87.3/97.7  |  70.3/86.4  | 75.2/93.4 |
> | CLIP-B32-Laion + TTA  + DN*  |  67.2/90.3   | 76.2/94.3 | 64.3/93.7  |  87.1/99.1    | 86.8/97.2  |  71.1/85.8  | 75.5/93.4 |
>
>
>
> ### 4, 61.0 is different from 63.2.
> The original CLIP paper maintains a list of potential prompts and validates the best prompt from the list using a validation dataset. In our paper, to avoid dependence on a validation dataset, we use a fixed prompt template “a photo of a {}” for all datasets, and this prompt template may not be the optimal one from the template list from the original CLIP paper.
>
> ### 5, Finetuning experiments on Imagenet.
> Finetuning CLIP on the image classification dataset will effectively turn the pre-trained CLIP into an image-classification model. This is different from the pre-training contrastive objective of CLIP. Since the goal of DN is to better align the contrastive objective with downstream uses, this extension is outside the scope of DN, and we believe it is more appropriate to be left for a future direction. We believe that our current extensive results over the tasks of cross-modal retrieval, zero-shot classification, image-caption metrics, and contrastive fine-tuning on image-caption datasets are sufficient in validating the role of DN to better align downstream use of vision-language models with their pre-training objectives.

---

> > ### Comment · Reviewer_UGnD · 2023-08-10
> > **Re: Response to rebuttal**
> >
> > Thanks to the authors for there response.
> >
> > ### Not experimenting with ensemble of prompt templates.
> > Even the proposed approach uses some samples from the validation set, so I don't really understand the argument of not using validation set for prompt ensembling or so.
> > It's a pretty common knowledge now that using multiple ensemble of templates does improve accuracy. While the exact 80 prompts in the CLIP paper might have been handpicked a bit, but authors should definitely try out there approach when using ensemble of prompts. This is especially because all there gains are simply superseded by prompt ensembled evaluation. Now if there approach doesn't give any gains over prompt ensembling, I do not see any practical significance.
> > Even for retrieval tasks (which authors are pushing as the main experiment of the paper), they should definitely try ensembling.
> >
> > ### Lack of various evaluation settings
> > The settings where the approach has to be used (for significant gains) seems to be too constrained. The authors argue that finetuning on image classification datasets is outside the scope of the paper. Even in zeroshot evaluation, the authors seem to argue that major improvements will come only in retrieval tasks. Can the authors share results when CLIP is finetuned for retrieval tasks, like on MSCOCO? Also not an important question, but can the approach be applied for multimodal captioning models as well to improve captioning performance like on CoCa and BLIP?
> >
> > ### Comparison to fewshot finetuning
> > The authors mention in the general response above that DN needs few test samples, and is therefore a fewshot and not zeroshot evaluation. Doesn't this, in turn, require the authors to add comparisons with a whole new host of baselines from fewshot finetuning of CLIP literature (CoOp: https://arxiv.org/pdf/2109.01134.pdf, TipAdapter: https://arxiv.org/pdf/2109.01134.pdf, CoCoOp: https://arxiv.org/abs/2203.05557)?

---

> > > ### Author Response · Authors · 2023-08-12
> > >
> > > ### 1, Use prompt ensembling
> > > Thanks for pointing this out. We have tried using DN with prompt ensembling using the 80 prompt templates used by the original CLIP paper as mentioned by the reviewer. We have tested the effects of ensembling prompt templates for cross-modal retrieval, classification, and image caption metrics. We found that DN still achieves consistent improvement for all tasks on top of other techniques like TTA and prompt ensembling, especially significant on cross-modal retrieval and image caption metrics.
> > >
> > > Cross-modal Retrieval with prompt ensembling(Top1/Top5/Top10)
> > > |  | Flickr30k I2T|   Flickr30k T2I       | MSCOCO I2T|      MSCOCO T2I   | avg |
> > > |-----------|---------------|------|--------|---------------|---|
> > > | CLIP-B32 + TTA |  82.5/96.4/98.4      | 63.6/87.2/92.2  | 54.0/77.7/85.4  |   31.9/57.1/68.1   | 58.0/79.6/86.1 |
> > > | CLIP-B32 + TTA + DN* |   84.3/97.0/98.5   | 66.1/88.2/93.1  | 54.1/77.8/85.6  |   33.6/59.0/69.4    | 59.5/80.5/86.7 |
> > >
> > > Classification with prompt ensembling(Top1/Top5)
> > > |  |imagenet1k |   Cifar100   |   SUN397     | Stanford Cars |  Caltech101   |   Flowers102     | avg |
> > > |-----------|---------------|------|--------|---------------|------|--------|---|
> > > | CLIP-B32 + TTA     | 64.3/89.6  |67.2/90.7 | 59.7/91.7 |  60.8/91.8 | 83.7/96.2 | 63.1/82.6  | 66.5/90.4  |
> > > | CLIP-B32 + TTA + DN*    | 64.3/89.6  | 68.2/90.7  | 60.1/91.8  |  61.0/91.8    | 83.7/96.3  | 63.3/82.9   | 66.8/90.5 |
> > >
> > > Image Caption Metrics with prompt ensembling(Kendall Tau)
> > > |  |Flickr8k-expert |   Flickr8k-cf       | THumb |     avg |
> > > |-----------|---------------|------|--------|---------------|
> > > | CLIP-B32 + TTA | 51.7      | 33.9   | 18.6 |  34.7    |
> > > | CLIP-B32 + TTA + DN* |  53.5    | 34.9  | 20.3 |  36.2     |
> > >
> > > ### 2, Lack of various evaluation settings.
> > > We respectively disagree that the evaluation setting of DN is limited. Since DN is proposed to align the downstream use better with the pre-training objective, it is in principle applicable to all cross-modal alignment tasks and is most effective in cases where the downstream task is most similar to the pre-training image-caption contrastive objective, including **both cross-modal retrieval and image caption metrics**. However, even in the classification task, DN also achieves a significant improvement, 0.9% to 1.4%, in most popular CLIP architectures pointed out by the reviewers. Furthermore, a larger gain (as large as 7% in classification task) is observed for other popular vision-language models like TCL and ALBEF pre-trained on smaller datasets, showing that DN is particularly effective in the case when there is a large distribution shift. In comparison, CoOp, TipAdapter, and CoCoOp only work in the few-shot setting and only apply to the classification task. Results of fine-tuning experiments on retrieval datasets have been provided in Figure 2 in the main paper. Since DN is similar to test-time adaptation but operates in a more relaxed setting than standard test-time adaptation methods as cited by reviewer 5Jmx, it is expected the larger the distribution shift, the larger the improvement. We have provided a detailed discussion with related test-time adaptation literature in response to reviewer 5Jmx and will include them in a revised version. For captioning models, it’s unclear how we can apply DN to improve captioning performances since DN is proposed for cross-modal alignment tasks where the similarity between text and images is measured. We would like to leave it for an interesting future direction.
> > >
> > >
> > > ### 3, Comparison with few-shot tuning
> > > Although we used the word few-shot, few-shot methods as cited by the reviewer cannot be applied in our setting. In our setting, DN can work with as few as **10 unlabeled samples** while the CoOp, TipAdapter, and CoCoOp models provided by the reviewer need at least a few samples per class, which adds up to **hundreds or even thousands of labeled samples**. We are happy to include their results as baselines in a revised version, but we have shown already that DN consistently works in a much relaxed setting compared to the few-shot learning literature.
> > >
> > > We are happy to address any further concerns.

---

> > > > ### Comment · Reviewer_UGnD · 2023-08-13
> > > > **Discussion**
> > > >
> > > > Thanks to the authors for the experiments and for resolving my concerns. I encourage the authors to incorporate these suggestions in the main paper. I am updating my rating to weak accept.

---

> > > > > ### Author Response · Authors · 2023-08-13
> > > > >
> > > > > Thanks so much for recognizing the value of our work and raising the score! We will include those experiment results in our revised version.

---

### Official Review · Reviewer_HYdZ · 2023-07-05

**Soundness:** 3 good
**Presentation:** 3 good
**Contribution:** 3 good
**Rating:** 4
**Confidence:** 4

**Summary:**

This paper addresses the problem of retrieval and classification accuracy using vision-language representation.  The authors claimed that there is a mismatch between training objectives and inference-time operation. Specifically, InfoNCE used negative information but test time only use positive similarity scores. The authors presented findings that rectifying such misalignment can boost performance consistently across a variety of downstream tasks. They argued that it is sufficient to use the first-order approximation of InfoNCE loss, i.e. just subtracting the mean in a test batch before doing dot product at test time, and called it Distribution Normalization (DN). They applied their test-time augmentation to a number of existing approaches such as CLIP, ALBEF, and TCL, and show certain improvements over the baselines on common classification and retrieval benchmarks such as MSCOCO and Flickr30K.

**Strengths:**

The paper is clearly written and the proposed approach and statement are simple and easy to understand. The experiments show consistent improvements over baselines that don't use test-time augmentation.
The authors applied their approach to some of the top performers and tested them on the various common benchmarks. They also provided a decent set of ablation studies on how test-time batch size affects performance and how this simple normalization help in the fine-tuning setting.

**Weaknesses:**

On line 110, the claim that 0.1 is small is not very convincing, since in practice, the value of the referred expressions is typically much smaller, especially after L2 normalization. Such oversimplification as this makes the final outcome, I.e. DN method, looks simplistic.

The approach in this paper required test-time augmentation, and when compared to other test-time augmentation approaches such as TTA of [54], the result was even inferior in some cases. Only when in combination with TTA, there was marginal improvements (over TTA[54])

**Questions:**

N/A

---

> ### Author Rebuttal · Authors · 2023-08-09
>
> ### 1, The claim that 0.1 is small is not convincing
> In our paper, we claim that the value of the referred expression is smaller than 0.1 (or even much smaller than 0.1 according to the reviewer) and therefore second and higher order terms are negligible. The result of this simplification is to make DN as simple as possible so that it is easy and efficient to implement. As provided in Appendix D.2, we have also shown that adding the second and higher order terms won’t make a big difference compared to our first order approximation:
> Part of Appendix D.2
> |  |Flickr8k-expert|    Flickr8k-cfI       |THumb|
> |-----------|---------------|------|--------|
> | CLIP+ DN |    54.3     | 35.4 |  23.5  |
> | CLIP+ DN (full) |    54.3     | 35.4|  23.4  |
> | TCL+ DN |    42.0     | 26.4 |  14.4  |
> | TCL+ DN (full) |    41.8     | 26.4 |  14.3  |
> | ALBEF+ DN |    34.8     | 21.8|  5.5  |
> | ALBEF+ DN (full) |     34.8     | 21.8|  5.5  |
> We would be happy to share more details if the reviewer can please explain more why this is a weakness.
>
> ### 2, The method is inferior to TTA. TTA + DN only has marginal improvement over TTA
> We have provided a general response on this issue and kindly ask the reviewer to refer to the “Performance Gain” general response for this concern.

---

### Official Review · Reviewer_jN4m · 2023-07-07

**Soundness:** 3 good
**Presentation:** 3 good
**Contribution:** 2 fair
**Rating:** 4
**Confidence:** 4

**Summary:**

This paper focus on the misalignment of training and testing of CLIP model. Specifically, CLIP is trained with an InfoNCE	loss containing both positive and negative samples, while tested lack negative samples information. They reveal that the common downstream practice of taking a dot product is only a zeroth-order approximation of the optimization goal, and propose distribution normalization to narrow the gap at test time considering both effectiveness and efficiency. Experimental results show some improvements on different datasets.

**Strengths:**

-This paper propose an interesting perspective for the difference of the optimization goal between train- and test-time.
-The analysis of InfoNCE Loss is easy to understand.
-The paper is well written, and easy to follow.


**Weaknesses:**

-There is no relevant introduction about test-time tasks in the first paragraph, and the necessity of performing this task at test-time is questionable.
-The proposed distribution normalization requires prior knowledge of the entire test set, which is often not met in practice, making it unable to handle sustained data.
-The performance gain is limited. Is it the limitation of the first-order approximation?


**Questions:**

see weakness.

**Limitations:**

The limitations are adequately addressed in the paper and seem reasonable.

---

> ### Author Rebuttal · Authors · 2023-08-09
>
> ### 1, The necessity of performing this task at test time is questionable.
> We are not sure about what “this task” refers to, and would be happy to give more explanations if the reviewer can please clarify this point.  However, just to clarify as best as we could understand the comment, DN is conducted to improve performance by aligning training and test time objectives, which we proved with strong test-time results. Reviewer’s comment on why this is necessary and how our approach is questionable is puzzling.
>
> ### 2, DN requires knowledge of the entire test, unable to handle sustained distribution
> We have provided a general response on this issue and kindly ask the reviewer to refer to the “Need for Extra Test Samples” general response for this concern.
>
> ### 3, The performance gain is limited
> We have provided a general response on this issue and kindly ask the reviewer to refer to the “Performance Gain” general response for this concern.

---

> > ### Comment · Reviewer_jN4m · 2023-08-21
> >
> > Thank the authors for their rebuttal and address my concerns, I decide to raise my score to borderline accept.

---

> > > ### Author Response · Authors · 2023-08-21
> > >
> > > We thank the reviewer for agreeing to raise the score! We wonder if the reviewer can also update the score in the system?

---

### Author Rebuttal · Authors · 2023-08-09

We appreciate the reviewers’ agreement on the efficiency and wide applicability of our proposed DN, and reviewer XtRE for recognizing the effectiveness of DN. There were several issues pointed out by multiple reviewers which we will address here in the general response.

### Performance Gain:
First of all, while reviewer HYdZ is right in saying that DN is comparable to TTA on average, we want to emphasize that DN is not mutually exclusive to TTA methods as they serve different purposes. DN aligns objectives between training and testing while TTA resolves issues caused by the instability of a single view of the image.  As such, they can work hand in hand together to produce strong results. For example, as shown in Section 4, DN can be combined with TTA techniques such that CLIP+TTA+DN consistently performs better than CLIP+TTA across all tasks.

More importantly, our absolute performance gain over baseline methods is comparable to the recently published [1, 2] works that reviewers XtRE and 5Jmx cited in their reviews. For cross-modal retrieval task top-1 retrieval accuracy, our CLIP+TTA+DN* achieves another 1.6% average performance boost over CLIP+TTA. The equivalent average performance boosts for classification and image captioning metrics are 0.9% and 1.6%. A similar improvement of CLIP+DN compared to CLIP is 2.2% for retrieval, 0.8% for classification and 1.3% for image caption metrics. In fact, when using the same CLIP architecture CLIP-B16 as in [1] and [2], the improvements are even as large as 1.4% top-1 accuracy for classification (see full results in the response to UGnD). In comparison, the average improvements of top-1 accuracy of [1] and [2] for zero-shot cross-dataset classification are 1.52% and 2.12% respectively compared to their baselines, but their method does not generalize to tasks beyond classification. In contrast, DN is widely applicable to a wide variety of important multi-modal alignment tasks including cross-modal retrieval, classification, and image captioning metric and at the same time achieve decent improvements across the board.

In addition, we would like to direct the reviewers’ attention to the more neglected experiments with TCL and ALBEF that achieve a much larger gain compared to CLIP (as large as 7% average top-1 improvement on classification). We surmise that this is because TCL and ALBEF are pre-trained on a smaller dataset compared to CLIP, which makes them more susceptible to larger distribution shifts in downstream applications. The relatively larger improvement from adding DN here should not be overlooked, as it shows that it can be particularly helpful when added to models with significant real-world distribution shifts such as specialized expert models pre-trained on a smaller dataset. This does not mean that DN will be ineffective for CLIP with a larger model architecture and pre-trained on a larger dataset. As presented in response to reviewer UGnD, DN brings a comparable if not larger performance boost to CLIP-B16, CLIP-L14, and CLIP pre-trained on the laion2B dataset.

### Need for Extra Test Samples:
There were also several comments about the need for test samples. We want to ensure that it is clear that we only need a very small number of test samples as shown in Table 4, not the entire test set as reviewer jN4m claimed. However, we do admit that these few test samples need to be of the same distribution as reviewer 5Jmx pointed out. But we don’t agree that this is impractical in many real-world applications as long as we collect these test samples close in time (e.g., trending images on social networks). This requirement is strictly the weakest among the test-time adaptation literature mentioned by reviewer 5Jmx. We also agree that since we need a small number of test samples, we should be clear that DN is few-shots rather than zero-shots (L175), and will refine related descriptions in the paper.


### Practicality:
Finally, DN is particularly valuable in light of the practical tradeoff between efficiency and accuracy. [1] requires doing test-time adaptation gradient updates for each single test sample and [2] requires iterating over hundreds of prompt templates in the prompt pool. Both methods create a considerable computational overhead. In contrast, DN is particularly simple to implement — we simply subtract the mean.


[1] Shu, Manli, et al. "Test-time prompt tuning for zero-shot generalization in vision-language models.", NeurIPS 2022.

[2] Allingham, James Urquhart, et al. "A Simple Zero-shot Prompt Weighting Technique to Improve Prompt Ensembling in Text-Image Models.", ICML 2023.

---

### Author Response · Authors · 2023-08-17
**Happy to address more concerns.**

We would like to thank reviewers jN4m and HyDz for reviewing our paper. As it is drawing close to the deadline of the discussion period, we would like to check in to see if our rebuttal addressed the concerns in the original review and if there are any further concerns after the rebuttal we can address to raise the rating of our paper. Please let us know!

---

### Decision · Program_Chairs · 2023-09-21

**Decision:**

Accept (poster)

**Comment:**

This paper proposes a simple method to improve zero-shot accuracy of CLIP style models. The method is well motivated by the info-nce loss used in contrastive learning. They have quite strong results and somewhat thorough evaluation (the inclusion of other test time training methods was particularly useful). Reviewers also agreed to increase the scores after the rebuttal. Thus